# Effects of Nutrient Source, Temperature, and Salinity on the Growth and Survival of Three Giant Clam Species (Tridacnidae)

**DOI:** 10.3390/ani14071054

**Published:** 2024-03-29

**Authors:** Rod Ping-Tsan Lee, Yu-Ru Lin, Chih-Yang Huang, Fan-Hua Nan

**Affiliations:** Department of Aquaculture, College of Life Science, National Taiwan Ocean University, Keelung City 202301, Taiwan; 20633003@mail.ntou.edu.tw (R.P.-T.L.); linyuru09@gmail.com (Y.-R.L.)

**Keywords:** climate change, giant clam, growth performance, nutrient, salinity, survival rate, temperature

## Abstract

**Simple Summary:**

This study addressed the pressing issue of declining giant clam populations due to environmental changes in their habitats. Our findings underscore the critical role of environmental factors in shaping clam populations and emphasize the importance of understanding these dynamics for effective conservation strategies. This study contributed valuable insights to the intricacies of giant clam ecology, offering a scientific foundation for informed conservation efforts aimed at preserving these vital marine species and their ecosystems.

**Abstract:**

The habitats of giant clams are undergoing environmental changes, and giant clam populations are declining. The present study was conducted to facilitate clam conservation. We conducted three 18-week trials to investigate the effects of nutrient, temperature, and salinity on the growth performance and survival rates (SRs) of juvenile *Tridacna noae*, adult *Tridacna crocea*, and subadult *Tridacna derasa*, respectively. Regarding nutrient sources, no significant differences were observed in shell length gain, specific growth rate, or SR between clams fed with *Chaetoceros muelleri* or commercial feed (hw nanotip) and those in a control group (juvenile phototrophs). Regarding temperature, clams cultivated at 27 °C exhibited significantly better growth performance and SR than did those cultivated at 19 °C or 31 °C (*p* < 0.05). By week 6, all clams in the 19 °C and 31 °C groups had died, indicating that suboptimal growth temperatures have severe adverse effects. Regarding salinity, clams cultivated at 34‰ exhibited significantly higher length gains and specific growth rates than did those cultivated at 20‰ or 25‰ (*p* < 0.05). SR was not significantly affected by salinity. Understanding how environmental factors affect giant clam populations may help researchers devise effective clam conservation strategies.

## 1. Introduction

Giant clams (Bivalvia: Tridacnidae) inhabit the shallow coral reefs scattered across the Indo-Pacific region [1,2]. Historically esteemed as prized fishery specimens, these creatures have vibrant mantles and have found their place in the wildlife ornament market [3]. Their adductor muscle and mantle serve culinary and ornamental purposes, respectively. However, rampant overfishing and habitat degradation have led to a marked decline in the population of giant clams [4,5,6]. Consequently, their trade is now regulated by Appendix II of the United Nations Convention on International Trade in Endangered Species of Wild Fauna and Flora.

The primary challenge faced by giant clams is climate change, which is a global threat to coral reef ecosystems [7,8,9]. Over the last 150 years, a consistent increase has been noted in global surface temperatures. Projections indicate a further surge of 2.6°C–4.8 °C in the 21st century. This upward trend may increase the frequency and potency of cyclones and rainfall, posing a risk of reduced salinity—a major stressor. Thus, global climate change simultaneously leads to increased temperatures and reduced salinity levels.

Environmental shifts induce a cascade of biological phenomena [8,10,11,12]. For energy, giant clams heavily rely on photosynthesis conducted by their symbiotic dinoflagellate partners (zooxanthellae; Symbiodiniaceae); filter-feeding serves as a secondary source of energy [8,13,14]. Notably, elevated temperatures (33 °C) and reduced salinity levels (10‰) substantially decelerate the growth of zooxanthellae, thereby increasing the risk of mortality in giant clams [13]. The loss of these crucial symbionts disrupts the ecological equilibrium of coral reef systems. For the preservation of giant clams, studies must be conducted to understand the effects of temperature and salinity on coral reefs. Accordingly, this study was conducted to investigate the effects of various nutrient sources, temperatures, and salinity on the growth performance and survival rate (SR) of three giant clam species, namely, *Tridacna noae* (juveniles), *Tridacna crocea* (adults), and *Tridacna derasa* (subadults), respectively.

## 2. Materials and Methods

### 2.1. Animal Source, Transportation, and Acclimation

A batch of juvenile *T. noae* (*n* = 180) was sourced from the Penghu Marine Biology Research Center in Taiwan. This batch comprised clams (aged 2 months, hosting endosymbionts) with an initial mean shell length (SL) of 1.52 ± 0.69 mm. The clams were meticulously packed in double-layered plastic bags with oxygen supply and airlifted to Yilan, Taiwan. Simultaneously, adult *T. crocea* clams (*n* = 18) and subadult *T. derasa* clams (*n* = 18) were imported (through Lamarck Marine Center, Taiwan) from Vietnam.

Upon their arrival, juvenile *T. noae* clams were introduced into designated rearing tanks. The tank base was outfitted with square-shaped bricks arranged in a grid formation (each grid measuring 3 cm × 3 cm), creating a surface area of 21 cm × 21 cm (comprising 49 bricks). Each brick was uniquely marked with an identification number, ensuring individual recognition and preventing duplication during specimen assessment. The survival status of each clam was confirmed upon their arrival at the Yilan greenhouse. The clams were acclimatized in artificial seawater for 2 weeks before experiments begun.

### 2.2. Experimental Water and Environment

An indoor aquaculture system was used to control growth conditions. The system comprised independent recirculating water systems for each experimental group. The system consisted of 18 glass tanks (60 cm × 35 cm × 30 cm; L × W × H), each with a water-holding capacity of 55 L. Each tank was equipped with a separate canister filter (2213; Eheim, Germany; flow rate: 440 L/h; filter media volume: 3 L) containing identical biomedia (2213-specific blue–white media; Eheim) and quartz balls (quartz balls; Eheim) to facilitate biomechanical filtration within a closed-loop system. The filtration process ensured the consistency of water quality and the removal of metabolic and excretory waste produced during the experimental period. Water temperature was maintained at 19 °C and 27 °C by using a chiller unit (TK150; 1/10 HP; Teco SRL, Canosa di Puglia, Italy) and at 31 °C by using a heating element (75 W; Eheim).

The artificial seawater was prepared with tap water containing a blend of sea salt (Fish Live, Taiwan) to ensure nutrient consistency and to eliminate pathogens and contaminants. The water mixture was prepared 24 h in advance, allowing sufficient time for complete dissolution and homogeneous mixing through aeration. The water was maintained at 27.0 °C ± 1.0 °C. Water quality parameters were set as follows: dissolved oxygen, 6 ± 1 mg/L; salinity, 20‰, 25‰, or 34‰; pH, 8.2 ± 0.2; and alkalinity, 8 ± 1. The concentrations of key elements in the water were as follows: calcium, 400 ± 50 ppm; magnesium, 1200 ± 100 ppm; and potassium, 380 ± 20 ppm. These concentrations were maintained to approximate or mirror the conditions in coastal seawater or areas inhabited by giant clams. Approximately 20% (11 L) of the tank water was changed every week to replenish the macro and trace elements consumed by the clams. Before water replacement, algae formed on the tank walls were scraped off to avoid contamination, which may influence the biota and experimental outcomes.

Light-emitting diode aquarium lights (parameters: 390–770 nm, 6000–6500 K, 12,500 lx, and 30 W; Fish Live, Taiwan) were horizontally arranged above the aquariums. To achieve consistent light intensity, uniform illumination was ensured by adjusting the height of the clams in the water column. Furthermore, the lights were strategically positioned to avoid abnormal heating and evaporation (factors that may affect the experimental outcomes), thereby maintaining consistency in environmental conditions, illumination duration, and temperature control.

### 2.3. Effects of Various Factors on Growth Performance and SR

#### 2.3.1. Effects of Nutrient on Juvenile *T. noae*

Juvenile *T. noae* clams (n = 180; initial mean SL, 1.52 ± 0.69 mm) were cultivated for 18 weeks under different nutrient conditions: photoautotroph (control, juvenile clams hosting symbiotic dinoflagellate algae), fed with *Chaetoceros muelleri* (Full-Algae Biotechnology, Keelung City, Taiwan) and with a commercial feed (hw nanotip; HW Wiegandt, Nordrhein-Westfalen, Germany). Three tanks (20 juveniles per tank) were maintained for each experimental group.

#### 2.3.2. Effects of Temperature on Adult *T. crocea*

Adult *T. crocea* clams (n = 18; initial mean SL, 55.89 ± 26.63 mm) were cultivated for 18 weeks at 19 °C, 27 °C, and 31 °C. Three tanks (two adults per tank) were maintained for each experimental group.

#### 2.3.3. Effects of Salinity on Subadult *T. derasa*

Subadult *T. derasa* clams (n = 18; initial mean SL, 47.58 ± 24.62 mm) were cultivated for 18 weeks at salinity levels of 20‰, 25‰, and 34‰. Three tanks (two subadults per tank) were maintained for each experimental group.

### 2.4. Measurement of Growth Performance and SR

#### 2.4.1. Length Gain

The rate of shell length (SL) growth was estimated by measuring the shell’s maximum front and rear dimensions. For this, the shell was placed on a flat surface and measurements were taken using a digital caliper (precision: 0.01 mm; Mitutoyo, Kanagawa, Japan). The growth rate was calculated every 2 weeks by using the following formula:$$Length\;gain\;\left(\%\right)=\frac{Final\;SL-Initial\;SL}{Initial\;SL}\times 100\%$$

#### 2.4.2. SR

SR was measured every 2 weeks by using the following formula:$$SR\;\left(\%\right)=\frac{Final\;clam\;count}{Initial\;clam\;count}\times 100\%$$

#### 2.4.3. Specific Growth Rate

Wet weight was measured every 2 weeks by using a portable balance (220 g × 0.001 g; SPX223; Ohaus, Parsippany, NJ, USA). The specific growth rate (SGR) was calculated using the following formula:$$SGR\;\left(\%\right)=\frac{[\ln final\;wet\;weight\;\left(g\right)-ln\;initial\;wet\;weight\;(g)]}{Time\;(days)}\times 100\%$$

### 2.5. Statistical Analysis

Statistical analyses were performed using SAS (version 9.0; SAS Institute, Cary, CA, USA). One-way analysis of variance was performed to assess biweekly changes in all parameters. Statistical significance (*p* < 0.05) was determined using Duncan’s new multiple range test. Data are presented using mean ± standard deviation values.

## 3. Results

### 3.1. Effects of Various Nutrient Sources on the Growth Performance and SR of Juvenile T. noae

At the end of the 18-week feeding experiment, the final mean SL was 7.85 ± 1.05 mm in the *Chaetoceros muelleri* group, 7.96 ± 1.09 mm in the hw nanotip group, and 6.96 ± 1.00 mm in the phototroph group (Table 1). The length gain was 400.69 ± 56.26% in the *Chaetoceros muelleri* group, 424.72 ± 54.18% in the hw nanotip group, and 359.08 ± 4.04% in the photoautotroph group (Table 1). Furthermore, the SGR was 1.27 ± 0.09% in the *Chaetoceros muelleri* group, 1.31 ± 0.08% in the hw nanotip group, and 1.21 ± 0.01% in the photoautotroph group (Table 1). No significant between-group differences were observed for any of the three aforementioned parameters (*p* > 0.05).

The SR values were 73 ± 5% in the *Chaetoceros muelleri* group, 66 ± 5% in the hw nanotip group, and 73 ± 5% in the photoautotroph group (Table 2). No significant between-group difference in SR was observed (*p* > 0.05).

### 3.2. Effects of Various Temperatures on the Growth Performance and SR of Adult T. crocea

In week 4, the final mean SL was 67.15 ± 5.25 mm in the 19 °C group, 80.22 ± 3.91 mm in the 27 °C group, and 68.99 ± 6.75 mm in the 31 °C group (Table 3). The length gain in week 4 was 0 ± 0% in the 19 °C group, 12.27 ± 2.52% in the 27 °C group, and 0 ± 0% in the 31 °C group. The length gain in the 27 °C group was significantly higher than that in the 19 °C and 31 °C groups (*p* < 0.05). Furthermore, the SGR was 0.08 ± 0.14% in the 19 °C group, 3.08 ± 0.34% in the 27 °C group, and 0 ± 0% in the 31 °C group. Significant differences were noted between the 27 °C and other groups in the three aforementioned parameters (*p* < 0.05).

In week 4, the SR was 33 ± 28% in the 19 °C group, 16 ± 28% in the 31 °C group, and 100% in the 27 °C group (Table 4). By week 6, all clams in the 19 °C and 31 °C groups had died, highlighting the detrimental effects of suboptimal temperatures on the SR of *T. crocea*. However, the 27 °C group exhibited a consistent SR (100%) throughout the experimental period. Significant differences were noted between the 27 °C and other groups in SR (*p* < 0.05).

### 3.3. Effects of Various Salinity Levels on the Growth Performance and SR of Subadult T. derasa

The final mean SL was 62.08 ± 10.16 mm in the 20‰ salinity group, 62.46 ± 12.51 mm in the 25‰ salinity group, and 65.17 ± 11.44 mm in the 34‰ salinity group (Table 5). The length gain was 0.67 ± 0.52% in the 20‰ group, 3.1 ± 0.23% in the 25‰ group, and 12.46 ± 2.5% in the 34‰ group (Table 5). Furthermore, the SGR was 1.3 ± 0.24% in the 20‰ group, 1.34 ± 0.38% in the 25‰ group, and 3.07 ± 0.31% in the 34‰ group. The three aforementioned parameters were significantly higher in the 34‰ group than in the other two groups (*p* < 0.05).

The SR was 100% for all experimental groups (Table 6). No significant between-group difference in SR (*p* > 0.05) was observed.

## 4. Discussion

### 4.1. Nutrient Sources

Nutrient availability, particularly in larval stages, is crucial for the growth performance and SR of giant clams. Studies have extensively explored the effects of feeding on the survival of bivalve larvae [15,16,17]. In the aquaculture of giant clams, various algal strains have been used to enhance the survival of clam larvae [18,19,20]. Our findings corroborate those of the study demonstrating that feeding different algal strains and yeast improved the SR of *Tridacna squamosa* larvae [17]. However, the specific effects of different nutrients on the growth of giant clams remain somewhat unclear. Thus, we investigated the effects of various nutrient sources on the growth performance and SR of juvenile *T. noae*. Although the final mean SL varied slightly across the groups, particularly between the nutrient-fed and photoautotroph groups, the differences were nonsignificant. Similarly, no significant between-group differences were observed in length gain or SGR. Moreover, the SR did not vary significantly among the groups, consistent with findings by Neo et al. [17].

The aforementioned findings suggest that the nutrient sources tested in our study do not substantially affect the growth performance and SR of juvenile *T. noae* growing under conditions similar to those of this study. Although the literature highlights the importance of feeding regimes, particularly with regard to larval development and survival, our study indicates that specific nutrient compositions do not significantly affect the growth and survival of juvenile giant clams. Nonetheless, further studies are needed to elucidate the potential interactive effects of different nutrient sources and environmental factors on the physiology and overall fitness of giant clam populations. Such studies would help us optimize aquaculture practices and preserve giant clam populations in the face of current environmental challenges.

### 4.2. Temperatures

Giant clams exhibit sensitivity to changes in water temperature, particularly because of their symbiotic relationship with zooxanthellae. Heat stress can disrupt this symbiotic relationship, leading to bleaching events [21]. We investigated the effects of various temperatures on the growth performance and SR of adult *T. crocea*. Temperature significantly influenced the SR of these clams. Pronounced reductions in SR were observed at both 19 °C and 31 °C compared with the findings at 27 °C. Zhou et al. have previously shown using *T. crocea* that heat stress results in changes in the expression of the apoptotic gene caspase-3 and a decline in clam–symbiont density, which potentially mediate temperature-induced mortality [22]. Furthermore, a comprehensive 14-year investigation into the impact of sea surface temperature (SST) anomalies on the population dynamics of *Tridacna maxima* within the Lakshadweep reefs revealed that when mean summer SST exceeded 30 °C, it instigated bleaching events in *T. maxima* [10]. This observation strongly implies that elevated temperatures exert adverse effects on the survival rates of giant clam populations. Moreover, elevated temperatures (33 °C) have been observed to diminish the larval survival rate in *Tridacna gigas* [23]. In the present study, we observed prominent effects of temperature on the growth of adult *T. crocea*. Clams cultivated at 27 °C exhibited superior growth performance than did those cultivated at 19 °C or 31 °C. This difference in growth outcomes indicates that temperature is a key environmental factor influencing the physiology and overall fitness of giant clams.

An essential consideration to underscore is the role of symbiotic zooxanthellae, serving as a primary nutrient source for giant clams. Warner et al. elucidated that elevated temperatures (32 °C and 34 °C) diminish the photosynthetic efficiency of zooxanthellae [24]. Consequently, this decrease raises the concern that under such conditions, zooxanthellae may not adequately supply nutrients to sustain giant clam populations. This insight suggests a potential mechanism by which temperature-induced stress could disrupt the vital nutritional relationship between zooxanthellae and giant clams.

Our study underscores the vulnerability of adult *T. crocea* to temperature variations, with 27 °C emerging as the optimal temperature for the growth and survival of *T. crocea*. These findings clarify the ecological dynamics of giant clams and emphasize the need for considering temperature fluctuations when designing conservation and management strategies for these iconic reef organisms. The molecular mechanisms underlying the temperature sensitivity of giant clams should be elucidated to facilitate the development of targeted conservation strategies and mitigate the effects of climate change on vulnerable marine ecosystems.

### 4.3. Salinity

Global climate change has led to a reduction in seawater salinity, such as extreme rainfall, and this reduction may affect the physiological functions and survival of marine bivalves [25,26,27,28]. Hyposalinity can induce osmotic stress in bivalves, potentially leading to mortality in extreme cases [29,30,31,32,33,34]. For instance, it has been observed that decreased salinities (18‰, 22‰, 26‰) have a detrimental effect on the fertilization success and early larval development of the giant clam *Tridacna gigas* [32], suggesting that intensified precipitation events could exacerbate the challenges faced by giant clam populations on the reef, potentially compromising their survival.

We investigated the response of subadult *T. derasa* to various salinity levels and found consistently high SRs across all levels (20‰, 25‰, and 34‰). Our findings indicate the robustness of subadult *T. derasa* to salinity variations. However, despite the high SR, subadult *T. derasa* exposed to hyposalinity exhibited reduced length gain and SGR compared with the parameters in clams maintained at relatively high levels of salinity. The observed growth retardation under hyposalinity conditions may be attributed to a decrease in respiration rates [35], which, in turn, reduces metabolic costs associated with maintenance and enhances the clams’ capacity to withstand environmental fluctuations [36]. Additionally, Maboloc et al. conducted a study in juvenile giant clams *Tridacna gigas* revealing that exposure to reduced salinity induced epithelial hyperplasia and hypertrophy in the clam’s ctenidial lamellae, characterized by closely packed cells exhibiting highly granulated nuclei with ruptured nuclear membranes. Moreover, structural alterations such as fusions and elongations of the branchial filaments with eroded frontal cilia were observed [33]. These findings not only shed light on the intricate physiological responses of giant clams to changes in salinity levels but also underscore the multifaceted impact of salinity fluctuations on the anatomical integrity and feeding behavior of these organisms. Further investigation into the mechanisms underlying these responses is essential for elucidating the adaptive strategies of giant clams in the face of changing environmental conditions.

In summary, our findings suggest that although subadult *T. derasa* demonstrate resilience to varying salinity levels, hyposalinity causes growth retardation. The physiological mechanisms underlying the response of giant clams to hyposalinity should be elucidated to inform conservation strategies amidst ongoing environmental changes.

## 5. Conclusions

Our study clarifies the growth performance and SR of giant clams under different environmental conditions. Nutrient sources may not significantly influence the growth and survival of juvenile giant clams. However, temperature strongly affects the physiology and overall fitness of adult clams. For *T. crocea*, 27 °C appears to be the optimal temperature for growth and survival. Although subadult *T. derasa* exhibit resilience to varying levels of salinity, hyposalinity conditions may lead to growth retardation. Future studies should investigate the interactive effects of nutrients, temperature, and salinity on giant clam populations to guide conservation and management strategies for overcoming current environmental challenges such as climate change.

## Figures and Tables

**Table 1 animals-14-01054-t001:** Effects of various nutrient sources on the growth performance and survival rate of juvenile *Tridacna noae.*

Groups	IL ^3^ (mm)	FL ^3^ (mm)	IW ^3^ (g)	FW ^3^ (g)	LG ^3^ (%)	SGR ^3^ (%)	SR ^3^ (%)
Control ^3^	1.52 ± 0.68 ^a,1,2^	6.96 ± 1.00 ^a,1,2^	1.12 ± 0.03 ^a,1,2^	2.33 ± 0.03 ^a,1,2^	359.08 ± 4.04 ^a,1,2^	1.21 ± 0.01 ^a,1,2^	40 ± 4.08 ^a,1,2^
*Chaetoceros muelleri*	1.53 ± 0.70 ^a,1,2^	7.85 ± 1.05 ^a,1,2^	1.23 ± 0.05 ^a,1,2^	2.50 ± 0.14 ^a,1,2^	400.69 ± 56.26 ^a,1,2^	1.27 ± 0.09 ^a,1,2^	40 ± 4.08 ^a,1,2^
HW nano tip	1.53 ± 0.70 ^a,1,2^	7.96 ± 1.09 ^a,1,2^	1.11 ± 0.02 ^a,1,2^	2.42 ± 0.10 ^a,1,2^	424.72 ± 54.18 ^a,1,2^	1.31 ± 0.08 ^a,1,2^	41 ± 0 ^a,1,2^

^1^ Data are presented in terms of the mean ± standard deviation values. ^2^ Mean values in the same column denoted by the letter a indicate equal significant differences. ^3^ IL, initial length; FL, final length; IW, initial weight; FW, final weight; LG, length gain; SGR, specific growth rate; SR, survival rate. Control: juvenile *T. noae* hosting symbiotic dinoflagellate algae.

**Table 2 animals-14-01054-t002:** Biweekly changes in the survival rate of juvenile *Tridacna noae* receiving nutrients from different sources.

	Weeks/SR(%)	2	4	6	8	10	12	14	16	18
Groups		Survival Rate (%)								
Control		76 ± 6 ^a,1,2^	61 ± 6 ^a,1,2^	46 ± 4 ^a,1,2^	43 ± 6 ^a,1,2^	41 ± 4 ^a,1,2^	40 ± 4 ^a,1,2^	40 ± 4 ^a,1,2^	40 ± 4 ^a,1,2^	40 ± 4 ^a,1,2^
*Chaetoceros muelleri*		66 ± 4 ^a,1,2^	68 ± 4 ^a,1,2^	58 ± 2 ^a,1,2^	51 ± 2 ^a,1,2^	46 ± 4 ^a,1,2^	41 ± 4 ^a,1,2^	40 ± 4 ^a,1,2^	40 ± 4 ^a,1,2^	40 ± 4 ^a,1,2^
HW nano tip		73 ± 6 ^a,1,2^	63 ± 2 ^a,1,2^	58 ± 2 ^a,1,2^	53 ± 2 ^a,1,2^	46 ± 2 ^a,1,2^	41 ± 2 ^a,1,2^	41 ± 2 ^a,1,2^	41 ± 2 ^a,1,2^	41 ± 0 ^a,1,2^

^1^ Data are presented in terms of the mean ± standard deviation values. ^2^ Mean values in the same column denoted by the letter a indicate equal significant differences.

**Table 3 animals-14-01054-t003:** Effects of temperature on the growth performance and survival rate of adult *T. crocea.*

Groups	IL ^3^ (mm)	FL ^3^ (mm)	IW ^3^ (g)	FW ^3^ (g)	LG ^3^ (%)	SGR ^3^ (%)	SR ^3^ (%)
19 °C	67.15 ± 5.25 ^a,1,2^	67.15 ± 5.25 ^b,1,2^	37.16 ± 8.52 ^a,1,2^	37.16 ± 8.52 ^b,1,2^	0 ± 0 ^b,1,2^	0.08 ± 0.14 ^b,1,2^	0 ^b,1,2^
27 °C	67.79 ± 3.15 ^a,1,2^	80.22 ± 3.91 ^a,1,2^	33.9 ± 5.16 ^a,1,2^	36.98 ± 5.48 ^a,1,2^	12.27 ± 2.52 ^a,1,2^	3.08 ± 0.34 ^a,1,2^	100 ± 0 ^a,1,2^
31 °C	68.99 ± 6.75 ^a,1,2^	68.99 ± 6.75 ^b,1,2^	39.33 ± 6.05 ^a,1,2^	39.33 ± 6.05 ^b,1,2^	0 ± 0 ^b,1,2^	0 ± 0 ^b,1,2^	0 ^b,1,2^

^1^ Data are presented in terms of the mean ± standard deviation values. ^2^ Mean values in the same column denoted by the letter a, b indicate significant differences. ^3^ IL, initial length; FL, final length; IW, initial weight; FW, final weight; LG, length gain; SGR, specific growth rate; SR, survival rate.

**Table 4 animals-14-01054-t004:** Biweekly changes in the survival rate of adult *T. crocea* exposed to different temperatures.

	Weeks/SR(%)	2	4	6	8	10	12	14	16	18
Groups		Survival Rate (%)								
19 °C		100 ± 0 ^a,1,2^	33 ± 28 ^b,1,2^	-	-	-	-	-	-	-
27 °C		100 ± 0 ^a,1,2^	100 ± 0 ^a,1,2^	100 ± 0	100 ± 0	100 ± 0	100 ± 0	100 ± 0	100 ± 0	100 ± 0
31 °C		100 ± 0 ^a,1,2^	16 ± 28 ^b,1,2^	-	-	-	-	-	-	-

^1^ Data are presented in terms of the mean ± standard deviation values. ^2^ Mean values in the same column denoted by the letter a, b indicate equal significant differences.

**Table 5 animals-14-01054-t005:** Effects of salinity on the growth performance and survival rate of subadult *T. derasa*.

Groups	IL ^3^ (mm)	FL ^3^ (mm)	IW ^3^ (g)	FW ^3^ (g)	LG ^3^ (%)	SGR ^3^ (%)	SR ^3^ (%)
20‰	61.41 ± 9.91 ^a,1,2^	62.08 ± 10.16 ^b,1,2^	32.9 ± 12.72 ^a,1,2^	34.2 ± 11.74 ^b,1,2^	0.67 ± 0.52 ^b,1,2^	1.3 ± 0.24 ^b,1,2^	100 ± 0 ^a,1,2^
25‰	59.36 ± 12.6 ^a,1,2^	62.46 ± 12.51 ^b,1,2^	38.6 ± 18.68 ^a,1,2^	39.93 ± 19.01 ^b,1,2^	3.1 ± 0.23 ^b,1,2^	1.34 ± 0.38 ^b,1,2^	100 ± 0 ^a,1,2^
34‰	52.71 ± 12.79 ^a,1,2^	65.17 ± 11.44 ^a,1,2^	32.66 ± 10.78 ^a,1,2^	35.73 ± 10.71 ^a,1,2^	12.46 ± 2.5 ^a,1,2^	3.07 ± 0.31 ^a,1,2^	100 ± 0 ^a,1,2^

^1^ Data are presented in terms of the mean ± standard deviation values. ^2^ Mean values in the same column denoted by the letter a, b indicate significant differences. ^3^ IL, initial length; FL, final length; IW, initial weight; FW, final weight; LG, length gain; SGR, specific growth rate; SR, survival rate.

**Table 6 animals-14-01054-t006:** Biweekly changes in the survival rate of subadult *T. derasa* exposed to different salinity levels.

	Weeks/SR(%)	2	4	6	8	10	12	14	16	18
Groups		Survival Rate (%)								
20‰		100 ± 0 ^a,1,2^	100 ± 0 ^a,1,2^	100 ± 0 ^a,1,2^	100 ± 0 ^a,1,2^	100 ± 0 ^a,1,2^	100 ± 0 ^a,1,2^	100 ± 0 ^a,1,2^	100 ± 0 ^a,1,2^	100 ± 0 ^a,1,2^
25‰		100 ± 0 ^a,1,2^	100 ± 0 ^a,1,2^	100 ± 0 ^a,1,2^	100 ± 0 ^a,1,2^	100 ± 0 ^a,1,2^	100 ± 0 ^a,1,2^	100 ± 0 ^a,1,2^	100 ± 0 ^a,1,2^	100 ± 0 ^a,1,2^
34‰		100 ± 0 ^a,1,2^	100 ± 0 ^a,1,2^	100 ± 0 ^a,1,2^	100 ± 0 ^a,1,2^	100 ± 0 ^a,1,2^	100 ± 0 ^a,1,2^	100 ± 0 ^a,1,2^	100 ± 0 ^a,1,2^	100 ± 0 ^a,1,2^

^1^ Data are presented in terms of the mean ± standard deviation values. ^2^ Mean values in the same column denoted by the letter a indicate equal significant differences.

## Data Availability

Data are contained within the article.

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
