# Peer review of "Effects of Nutrient Source, Temperature, and Salinity on the Growth and Survival of Three Giant Clam Species (Tridacnidae)"

_animals, 2024, doi:10.3390/ani14071054_

Round 1

Reviewer 1 Report

Comments and Suggestions for Authors

The manuscript is interesting experimental study devoted to charismatic clam species of genus Tridacna inhabiting the shallow coral reefs. It is concluded that temperature is a key environmental factor influencing the physiology and overall fitness of giant clams. Also, the clams growth is negatively affected by low salinity. At the same time, nutrient conditions do not significantly affect the growth and survival of juvenile giant clams. The text is clear and easily understandable. While the manuscript provides interesting results, there are a few minor concerns that need to be addressed before publication.

The first is design of experiments. The authors used three giant clam species, namely, Tridacna noae (juveniles), Tridacna crocea (adults), and Tridacna derasa (subadults). Effects of nutrient were tested on juveniles of Tridacna noae. Other two species (and ages) were used to study effects of temperature and salinity. Why were different species and ages used in different experimental series? It is more logical to study one species or age group (to compare different species).

L. 130 “The rate of SL growth (mm/month) was estimated…”. However, in equation at L. 134, “SL growth rate” is nondimensional variable. Moreover, growth rate is not mentioned anywhere in the text of Results, where this (?) variable is noted as “length gain”. So, please clarify. Equation (L. 134) is, in fact, one of the way for calculation of specific growth of shell length. I suggest you should present absolute (mm/day and mg/day) as well as specific (equation at L. 145) growth rates for length and weight.

Reviewer 2 Report

Comments and Suggestions for Authors

The manuscript reads well.  There are a few items to check:

Line 43 Reference 9 does not mention climate change in the accepted sense. It is dealing with historical seasonal changes
Line 64 “comprised clamps” do you mean “clams”? see also "clamps" in line 65 and 69.
Methods 2.1 were the clams acclimatised in artificial seawater as in Line 89 or in seawater from another source?
Line 255-256. There was no measurement of caspase-3 or clam-symbiont density in this study. To make it clear to readers that this information is inferred from published data I suggest that you change the sentence, perhaps to “Zhou et al. [23] have previously shown using T. crocea that heat stress results in  changes in the expression of the apoptotic gene caspase-3 and a decline in clam–symbiont density, which potentially mediate temperature-induced mortality.”

One disadvantage to the current study is that the temperature and salinity parameters were fixed to one value. This is not the case in the wild fishery lagoons where temperature and salinity vary on a diurnal basis and this should be noted in the discussion. Your reference 9 shows this as do papers such as  "van Haren H. Pacific shallow lagoon high-resolution temperature observations. arXiv preprint arXiv:1910.03451. 2019 Oct 8."

Comments on the Quality of English Language

no comments to make

Reviewer 3 Report

Comments and Suggestions for Authors

Dear authors,

I just read you MS assessing responses of giant clams to different physicochemical conditions. The results are quite interesting and valuable, but the discussion needs to be improved (see details below). I also made a suggestion for the M&M section. These comments are listed below and hope they are useful to the MS improvement.

Line 109 and on: here I suggest you to explain why each species was used to assess the effect of a factor. Why not using a single species to assess all factors, or instead, assessing all factors for the three species?

Lines 222-285 (whole Discussion section): The authors need to discuss the results comparing them to other tropical bivalves and other sensitive organisms from tropical reefs, such as corals, sponges, and ascidians. They also need to discuss better the role zooxanthellae and how these organisms cope with nutrient, temperature and salinity, in order to help you to explain the results.
